# Physical Activity Levels and Women’s Mental Health After COVID-19 Infection

**DOI:** 10.3390/healthcare12232346

**Published:** 2024-11-24

**Authors:** Miloš Stamenković, Saša Pantelić, Saša Bubanj, Emilija Petković, Nikola Aksović, Adem Preljević, Bojan Bjelica, Tatiana Dobrescu, Adina-Camelia Şlicaru

**Affiliations:** 1Faculty of Sport and Physical Education, University of Niš, 18000 Niš, Serbia; kineziologija92@gmail.com (M.S.); spantelic2002@yahoo.com (S.P.); petkovicemilija@yahoo.com (E.P.); 2Faculty of Sport and Physical Education, University of Priština-Kosovska Mitrovica, 38218 Leposavić, Serbia; kokir87np@gmail.com; 3Department of Biomedical Sciences, State University of Novi Pazar, 36300 Novi Pazar, Serbia; apreljevic@np.ac.rs; 4Faculty of Physical Education and Sports, University of East Sarajevo, 71126 Lukavica, Bosnia and Herzegovina; vipbjelica@gmail.com; 5Department of Physical Education and Sport Performance, Vasile Alecsandri University, 600115 Bacau, Romania; slicaruadinacamelia@ub.ro

**Keywords:** COVID-19 infection, mental health, physical activity, women, levels

## Abstract

Background: The aim of this research is to investigate the associations between physical activity and mental health parameters (depression, anxiety, and stress) among women who have recovered from COVID-19; Methods: This research involved two measurements: the initial test, conducted 2-to-4 weeks post-COVID infection, and the final test, performed 14-to-16 weeks after the virus’s activity. The sample consisted of women (*n* = 190) aged 20 to 60 (47.60 ± 11.1, 47.60 ± 11.1, mean ± Std.Dev.) who were infected with COVID-19. To assess the level of physical activity, a longer version of the IPAQ questionnaire was used. Self-assessment of mental health was determined by a longer version of the DASS questionnaire; Results: The t-test analysis revealed significant differences in mental health and physical activity levels between the initial and final measurements. After three months, subjects showed lower mental health scores (indicating improvement) and higher Metabolic Equivalent of Task (MET) values across all physical activity domains, with moderate physical activity showing the greatest increase. The regression analysis showed that at the initial measurement, there was no statistically significant association of physical activity with mental health parameters. Three months after the initial measurement, regression analysis showed that there was a statistically significant association of physical activity with anxiety (*F* = 3.97; *p* = 0.000) and depression (*F* = 3.34; *p* = 0.001) but not with stress (*F* = 1.67; *p* = 0.106); Conclusions: This research revealed that higher levels of physical activity improved mental health in post-COVID-19 women, with varying effects on anxiety and depression depending on the activity domain.

## 1. Introduction

The Severe Acute Respiratory Syndrome Coronavirus 2 (COVID-19) rapidly became a global health crisis following the first recorded case in Wuhan, China, due to the virus’s swift transmission, which affected 146 countries by mid-March 2020 [1,2,3]. Following the World Health Organization’s declaration of a global pandemic in March 2020, governments implemented quarantine and social distancing measures to mitigate viral spread [4]. While necessary for infection control, these measures coincided with a rise in mental health challenges (depression, anxiety, and stress levels) across populations [5,6,7,8,9], which may be attributed to a variety of factors, including fear of illness, concerns for personal and family safety, and changes in daily life [10]. Research indicates that women were especially affected, facing disproportionate increases in mental health concerns compared to men [11,12].

In addition to mental health impacts, the pandemic’s lockdown measures resulted in reduced physical activity levels globally [13,14], further compounding mental health issues as physical activity is known to play an essential role in psychological well-being [15].

Evidence shows that physical activity can mitigate the negative mental health effects of the pandemic [16,17]. For instance, Hamer and associates [18] found that physical activity of any intensity reduces the risk of psychological stress, and Junior and associates [19] observed that even irregular physical activity is linked to lower symptoms of depression, anxiety, and stress, as assessed by the Depression, Anxiety, and Stress Scale (DASS).

Despite these benefits, research has shown that women generally engage in lower levels of physical activity than men, with a preference for moderate-intensity activities, often performed at home [20,21,22,23,24,25]. Moderate-intensity physical activity has well-documented benefits for mental health [26,27,28], including reducing fears related to COVID-19 infection [29].

However, there is limited research focused on the role of physical activity in supporting mental health specifically among women recovering from COVID-19. COVID-19 recovery poses unique mental health challenges, potentially influenced by gender-specific factors, which may affect psychological recovery and well-being [30].

Understanding the association between physical activity and mental health outcomes in this population is crucial, as exercise could offer a potential mechanism to alleviate post-recovery anxiety, depression, and stress in women.

Therefore, the aim of this research is to investigate the associations between physical activity and mental health parameters (depression, anxiety, and stress) in women who are recovering from COVID-19.

We hypothesized that higher levels of physical activity will be significantly associated with improved mental health parameters (depression, anxiety, and stress) at the final measurement following recovery from COVID-19.

## 2. Materials and Methods

### 2.1. Sample of Participants

Out of 230 working women infected with COVID-19 and selected for this study using random sampling, 40 either did not meet the criteria for participation or did not participate in the final measurement; therefore, the final sample consisted of participants (*n* = 190) aged between 20 and 60 (47.60 ± 11.1, mean ± Std.Dev.).

Sample size was estimated using the pwr.t.test() function from the *pwr* library in the R statistical software v4.4.0. Mean values and standard deviations were preliminarily evaluated from two samples: one consisting of 25 women infected with the coronavirus, and the second of 25 women who were recovering for three months following COVID-19 hospitalization. Based on these values, the minimum sample size was determined to be 117, using standard settings (alpha = 0.05, power = 0.8).

Inclusion and exclusion criteria were set for the subjects in the research.

Inclusion criteria were as follows:subjects diagnosed with the coronavirus;subjects who did not have severe symptoms and were not on ventilators;subjects treated at home;subjects hospitalized for up to seven days;subjects who were out of the hospital or self-isolation for no more than one month;subjects without any mental illnesses requiring specific medication.

The exclusion criteria were as follows:subjects not diagnosed with the coronavirus;subjects who had severe clinical symptoms and were on ventilators;subjects not treated at home;subjects hospitalized for more than seven days;subjects who were out of the hospital or self-isolation for more than one month;subjects with specific mental illnesses requiring medication.

These criteria collectively aimed to include a broad range of mild-to-moderate cases, allowing for a comprehensive view of recovery patterns among non-severe cases.

Before starting the test, the subjects were informed in detail about the benefits and consequences of the test and research participation. Each subject consented to voluntary participation in the research before the program started. Since it was not possible to obtain signed consent from all participants in person, some participants were sent a consent form for the study via email in a Word document. Thanks to the directors of the General Hospital Leskovac, health centers, clinics, and municipalities, we were able to obtain participants’ phone numbers, email addresses, and other relevant information for this study. Any subject could leave the testing process at any time.

The questionnaire was conducted by phone interview. All questionnaires were marked with a code, not only for easier identification of subjects from the initial and final measurements but also to compare their initial results with the final ones. All personal data of the subjects were protected by the researchers, and in no case were they to be misused. The subjects were completely familiar with the procedures of this research and the need for retesting three months after the initial measurement. The research was conducted in accordance with the Declaration of Helsinki and Recommendations for Scientific Research Involving Human Subjects [31]. In addition, the research was approved by the Institutional Review Board of the University of Niš as well as the Ethics Committees of the Health Center Leskovac and the General Hospital Leskovac.

The research was conducted in the Jablanica District in Southern Serbia. Participants included employees from the Leskovac hospital, several health centers, medical stations, and municipalities throughout the Jablanica District (Leskovac, Medveđa, Vučje, Grdelica, and Lebane).

### 2.2. Data Collection

This longitudinal study with two measurement points (initial and final tests) was classified as a pre-test–post-test study. The initial measurement was carried out after the subjects had the coronavirus infection. Depending on the severity of the clinical picture and the duration of the virus’s activity, the initial testing was carried out 2-to-4 weeks after the virus’s activity. The final measurement was carried out in the 14th to the 16th week after the virus activity. The research was conducted between February and May 2022.

### 2.3. Sample of Measuring Instruments

#### 2.3.1. Physical Activity

The level of physical activity was assessed using the longer version of the International Physical Activity Questionnaire (IPAQ), valid for scientific use [32]. The IPAQ questionnaire was in Serbian. The validity of the Serbian version of the IPAQ was confirmed in a study by Milanović and associates [33]. The IPAQ questionnaire allows for the assessment of physical activity levels across various domains, including work, transport, home, leisure time, and walking. It also evaluates different intensity levels, such as moderate and vigorous physical activity as well as overall physical activity. The results of the scores are presented as MET-minutes/week. To obtain these numerical values, participants were asked to select, from the questionnaire fields, the total activity expressed in minutes per day, as well as the number of days per week that the specified activity was performed. These data were then multiplied by the so-called MET coefficients, which indicate the intensity of physical activity. The MET coefficient was calculated for each type of activity. For example, all types of walking were included, and an average MET value was calculated based on them. The same procedure was applied to moderate-intensity physical activity and high-intensity physical activity.

#### 2.3.2. Mental Health

Mental health was assessed using the longer version of the DASS questionnaire. The strong metric properties of the DASS were confirmed in research by Lovibond and Lovibond [34]. The DASS was translated into Serbian and adapted by Zoran Protulipac [35]. The DASS (Depression, Anxiety, and Stress Scale) consists of three self-report scales with 42 items to measure depression, anxiety, and stress. Subjects rate each item on a four-point scale (0 = no symptoms, 1 = mild symptoms, 2 = moderate symptoms, 3 = severe symptoms). Each scale contains 14 items grouped into subscales, assessing specific symptoms. The depression scale includes symptoms like hopelessness and anhedonia; the anxiety scale assesses agitation and anxious feelings; the stress scale measures irritability and difficulty relaxing. Scores for each scale were calculated by summing the item scores.

### 2.4. Data Analysis

Data processing employed the statistical program SPSS (v23.0, SPSS Inc., Chicago, IL, USA). The normality of the data distribution was tested using the Kolmogorov–Smirnov test. For all of the testing data, basic statistical parameters were calculated: arithmetic mean and standard deviation. The paired samples t-test was applied for comparative analysis of the mean values between the initial and final measurements. In order to establish the influence level of physical activity on the parameters of mental health in the initial and final measurements, a linear regression analysis was used. A *p*-value of <0.05 was set as the level of statistical significance for all statistical procedures.

## 3. Results

The sample included a diverse range of sociodemographic characteristics (Table 1). In terms of education, most participants had completed high school (47.4%) or faculty (48.4%), with a smaller percentage holding an MSc/Ph.D. (2.6%) or only elementary school education (1.6%). Regarding employment status, the majority were employees (94.7%), with a small percentage being students (3.1%), unemployed (0.5%), or pensioners (1.6%). In terms of smoking behavior, the majority of participants were non-smokers (71.1%), while 28.9% were smokers. Regarding marital status, 75.8% were married, and 24.2% were unmarried. Lastly, the majority lived in cities (73.3%), with 26.3% residing in villages.

The paired samples t-test analysis revealed significant improvements in the mental health parameters between the initial and final measurements (Table 2). Specifically, the subjects showed reduced depression (9.66 ± 9.46 to 8.63 ± 8.47), anxiety (10.86 ± 8.81 to 9.53 ± 8.21), and stress (15.28 ± 9.36 to 14.47 ± 8.61), indicating enhanced mental health after three months. Cohen’s d indicates that while there are statistically significant differences between the initial and final measurements for all three mental health parameters, the effect sizes are small, suggesting that the magnitude of the change is modest.

In terms of physical activity (Table 3), all measured domains exhibited increased MET values, with notable improvements in overall physical activity (3664.70 ± 1602.46 to 4118.53 ± 1422.25) and moderate physical activity (1996.56 ± 996.76 to 2237.88 ± 1092.45). This indicates a general increase in physical activity levels among the participants. Effect sizes (Cohen’s d) show that the physical activity levels across various domains increased modestly from the initial to final measurements, with small to medium effects.

The regression analysis indicated no significant linear relationship between physical activity levels and mental health variables (depression, anxiety, and stress) at the initial measurement (Table 4). The multiple correlation coefficients (*R*) demonstrated that the predictors did not significantly influence mental health, as evidenced by the low *R²* values and no statistically significant levels (all *p* > 0.05).

The regression analysis revealed a significant linear relationship between physical activity levels and both anxiety (*R* = 0.387, *R*² = 0.150, *p* < 0.001) and depression (*R* = 0.359, *R*² = 0.129, *p* < 0.001) in subjects who had recovered from COVID-19 at the final measurement (Table 5); however, no significant relationship was found between physical activity and stress (*R* = 0.263, *R*² = 0.069, *p* = 0.106). The predictor variables showing the most substantial association with anxiety included overall physical activity (*β* = −19.39), moderate physical activity (*β* = −14.55), and intense physical activity (*β* = −10.50). For depression, the key predictors were overall physical activity (*β* = −15.70) and moderate physical activity (*β* = −11.69).

## 4. Discussion

The analysis of the descriptive statistics and t-tests revealed a difference between initial and final measurements in both physical activity and mental health parameters. Physical activity parameters showed higher values at the final measurement, while mental health parameters exhibited lower values, indicating an improvement. This suggests that higher levels of physical activity are associated with better mental health, as supported by Mendez-Gimenez and associates [16] who found that an MET value over 3000 lowers the risk of depressive symptoms. Physical activity is thus an important indicator of health, improving both physical and mental well-being [36,37,38].

The initial measurement regression analysis showed no significant association of physical activity with mental health. It should be noted that the IPAQ includes eight different domains, meaning that each domain affects anxiety, depression, and stress differently. Had we investigated only one domain, such as physical activity during leisure time, we might have obtained different results at the initial measurement. Leisure time is managed by the individual, which means that if they are motivated to be physically active for their own health, it is natural that they will enjoy such activities and experience improvements in mental health by neutralizing the negative effects of stress, anxiety, and depression on psychological and physiological homeostasis. In addition, the initial measurement was conducted only one month after recovery from an infection caused by COVID-19.

However, if we compare the MET values between the initial and final measurements (Table 3), we can conclude that the MET value is higher at the final measurement compared to the initial one. At the initial measurement, the total MET value of physical activity was 3664, while at the final measurement, it was 4118 MET. These data suggest that higher MET values for total physical activity are associated with improved mental health.

Specifically, walking, moderate physical activity, high-intensity physical activity, total physical activity, and physical activity at home were significant predictors of anxiety reduction. Previous studies have shown that physical activity while walking [39], moderate physical activity [40], and high-intensity physical activity [41] can reduce anxiety symptoms, which is in agreement with our study.

Additionally, our study identified physical activity at home as a significant factor, especially for women, aligning with research showing women are more active at home [24,25]. Cecchini and associates [28] as well as Popov and associates [39] also found that physical activity alleviates anxiety and depression.

Popov and associates [39] studied 680 adults during social isolation and demonstrated that physical activity, as a moderator, reduces anxiety. Similarly, our study found that walking, moderate physical activity, intense physical activity, total physical activity, and physical activity at home are significantly associated with reduced depression. Prior research has also confirmed the positive impact of these types of physical activity on depression [27,40,41].

At the final measurement, it was determined that physical activity is associated, at a statistically significant level, with anxiety and depression but not stress (Table 5). We believe that stress as a variable fluctuates, meaning it is always present regardless of a person’s physical activity level [42]. Simply put, individuals can experience stress due to family or work-related reasons, health issues, or negative emotions, which may explain the lack of a statistically significant association of physical activity with stress.

However, analyzing the changes in mental health parameters between the initial and final measurements reveals that stress levels were indeed lower at the final measurement compared to the initial one.

This finding supports the idea that some form and level of physical activity can help counteract the negative effects of stress. It is essential to remember that stress is always present, and its manifestation depends on numerous factors beyond physical activity alone.

Moderate physical activity (2237 MET) had the highest MET value at the final measurement (Table 2), suggesting a strong association between moderate physical activity and better mental health in women. Research consistently supports the positive effects of moderate physical activity on mental health [27]. Ghorbani and associates [43] also confirmed these findings. Additionally, our study indicated that moderate physical activity is common among women, which aligns with previous research [22].

Although high-intensity physical activity has been found to be more effective in reducing anxiety, depression, and stress [41], the benefits of moderate physical activity were also confirmed. Borrega-Mouguinho and associates [44] concluded that even moderate PA reduces symptoms of anxiety, depression, and stress, consistent with our results. Moreover, other studies highlight the importance of regular physical activity, regardless of intensity, for improving mental health [18].

## 5. Conclusions

The purpose of this study was to investigate the associations of different levels and domains of physical activity with the mental health of women who had recovered from COVID-19. Using standardized questionnaires to assess both mental health and physical activity, we found that higher levels of physical activity were associated with better mental health outcomes, specifically in reducing depression and anxiety. Moreover, the findings revealed that different domains of physical activity are associated with mental health parameters in distinct ways, highlighting the complexity of post-COVID recovery. This study adds to the existing body of research by offering gender-specific insights and emphasizing the importance of continued research in this area. Future studies should focus on longitudinal designs and use more objective measurement tools to provide more robust data on the long-term effects of physical activity on mental health after COVID-19.

## 6. Advantages and Shortcomings of This Study

This research contributed gender-specific insights, which have often been underrepresented in prior studies, on physical activity and mental health for women post-COVID.

While previous studies may have demonstrated a general link between physical activity and mental health, this study provided specific insights into post-COVID recovery in women, offering new perspectives on the role of moderate physical activity.

It is important to highlight the main methodological limitations of this research. Instead of using more valid and objective measurement instruments—such as an accelerometer—to determine the participants’ actual levels of physical activity, we used a questionnaire and telephone interviews to assess the levels of physical activity, which means caution is needed when interpreting the results due to the subjective nature of participants’ perceptions. In addition, we only investigated one district and did not consider other districts in Southern Serbia.

## Figures and Tables

**Table 1 healthcare-12-02346-t001:** Sociodemographic characteristics of the subjects (*n* = 190).

Sociodemographic Characteristics				
Education	Elementary school	High school	Faculty	MSc/Ph.D.
	3 (1.6%)	90 (47.4%)	92 (48.4%)	5 (2.6%)
Status	Student	Employee	Unemployed	Pensioner
	6 (3.1%)	180 (94.7%)	1 (0.5%)	3 (1.6%)
Health status	Smoker		Non-smoker	
	55 (28.9%)		135 (71.1%)	
Marital status	Married		Unmarried	
	144 (75.8%)		46 (24.2%)	
Place of residence	Village		City	
	50 (26.3%)		140 (73.3%)	

**Table 2 healthcare-12-02346-t002:** Descriptive statistics and comparative analysis of the paired samples *t*-test of mean values of mental health between the initial and final measurements of the subjects (*n* = 190).

Initial Measurement (mean ± SD)		Final Measurement (mean ± SD)		*t*	*p*	Cohen’s *d*
Depression	9.66 ± 9.46	Depression	8.63 ± 8.47	19.98	<0.001	0.11
Anxiety	10.86 ± 8.81	Anxiety	9.53 ± 8.21	23.47	<0.001	0.16
Stress	15.28 ± 9.36	Stress	14.47 ± 8.61	32.48	<0.001	0.09

Legend: Mean ± SD—mean value and standard deviation; *t*—t-statistic; *p*—statistical significance. The obtained values of mental health parameters are based on the DASS questionnaire.

**Table 3 healthcare-12-02346-t003:** Descriptive statistics and comparative analysis of the paired-sample *t*-tests of mean values of physical activity between the initial and final measurements of the subjects (*n* = 190).

Initial Measurement (mean ± SD)		Final Measurement (mean ± SD)		*t*	*p*	Cohen’s *d*
PA at work	1218.13 ± 1034.34	PA at work	1274.01 ± 871.27	27.31	<0.001	−0.06
PA in transport	374.80 ± 332.20	PA in transport	428.20 ± 385.82	19.88	<0.001	−0.15
PA at home	1189.36 ± 780.95	PA at home	1461.55 ± 873.33	14.54	<0.001	−0.33
PA in spare time	864.90 ± 725.11	PA in spare time	921.62 ± 749.60	12.75	<0.001	−0.08
PA while walking	950.63 ± 528.82	PA while walking	1061.69 ± 576.65	35.59	<0.001	−0.20
Moderate PA	1996.56 ± 996.76	Moderate PA	2237.88 ± 1092.45	39.56	<0.001	−0.22
High-intensity PA	717.51 ± 766.72	High-intensity PA	818.96 ± 755.21	19.81	<0.001	−0.13
Overall PA	3664.70 ± 1602.46	Overall PA	4118.53 ± 1422.25	49.97	<0.001	−0.30

Legend: mean ± SD—mean value and standard deviation; PA—physical activity expressed in MET; *t*—t-statistic; *p*—statistical significance.

**Table 4 healthcare-12-02346-t004:** Regression analysis of the association of physical activity with anxiety, depression, and stress at the initial measurement.

Initial Measurement	Depression		Anxiety		Stress	
Variables	*β*	*p*	*β*	*p*	*β*	*P*
PA at work	0.10	0.814	−0.13	0.769	−0.12	0.780
PA in transport	−0.06	0.686	−0.17	0.272	−0.07	0.660
PA at home	0.02	0.941	−0.04	0.893	−0.01	0.989
PA in spare time	0.14	0.642	−0.06	0.834	−0.01	0.981
PA while walking	−1.52	0.492	−1.24	0.567	0.89	0.688
Moderate PA	−2.57	0.533	−2.24	0.582	1.76	0.672
High-intensity PA	−2.05	0.530	−1.62	0.615	−1.53	0.643
Overall PA	4.01	0.542	3.78	0.559	−2.78	0.674
Depression *R* = 0.197, *R*^2^ = 0.039, *F* = 0.93, *p* = 0.507						
Anxiety *R* = 0.262, *R*^2^ = 0.069, *F* = 1.66, *p* = 0.109						
Stress *R* = 0.143, *R*^2^ = 0.020, *F* = 0.47, *p* = 0.875						

Legend: *β*—standardized beta coefficient; *p*—statistical significance; PA—physical activity; *R*—simple correlation; *R*^2^—partial coefficient of determination; *F*—model’s significance.

**Table 5 healthcare-12-02346-t005:** Regression analysis of the association of physical activity with anxiety, depression, and stress at the final measurement.

Final Measurement	Depression		Anxiety		Stress	
Variables	*Β*	*p*	*β*	*p*	*β*	*P*
PA at work	−0.27	0.300	−0.13	0.769	−0.16	0.546
PA in transport	−0.38	0.702	−0.18	0.272	−0.15	0.242
PA at home	−2.39	0.018	−0.61	0.027	−0.44	0.124
PA in spare time	−1.10	0.272	−0.17	0.834	−0.06	0.789
PA while walking	−6.50	0.007	−7.98	0.001	−2.90	0.237
Moderate PA	−11.69	0.009	−14,55	0.001	−5.02	0.270
High-intensity PA	−8.42	0.008	−10.50	0.001	−3.70	0.251
Overall PA	−15.70	0.006	−19.39	0.001	−6.91	0.237
Depression *R* = 0.359, *R*^2^ = 0.129, *F* = 3.34, *p* = 0.001						
Anxiety *R* = 0.387, *R*^2^ = 0.150, *F* = 3.97, *p* =< 0.001						
Stress *R* = 0.263, *R*^2^ = 0.069, *F* = 1.67, *p* = 0.106						

Legend: *β*—standardized beta coefficient; *p*—statistical significance; PA—physical activity; *R*—simple correlation; *R*^2^—partial coefficient of determination; *F*—model’s significance.

## Data Availability

The raw data supporting the conclusions of this article will be made available by the authors on request.

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
