# Peer review of "Physical Activity Levels and Women’s Mental Health After COVID-19 Infection"

_healthcare, 2024, doi:10.3390/healthcare12232346_

Round 1

Reviewer 1 Report (Previous Reviewer 2)

Comments and Suggestions for Authors

I wish you success in your future research.

Author Response

Dear Reviewer,

Thank you very much for taking the time to review this manuscript. 

Please find the corresponding revisions/corrections highlighted in the re-submitted file.

Kind regards,

The authors

Reviewer 2 Report (New Reviewer)

Comments and Suggestions for Authors

Thank you for the opportunity to review this paper. Here are my comments:

  1. Materials and Methods:

    • Sample Information: The sample data is confusing. It’s mentioned that the population from which the sample was drawn consists of 190 women, but the abstract also states that the sample included 190 women. Then, it mentions 25 women in the first and 25 women in the second testing. It’s necessary to clarify and specify:
      • From which population the sample was drawn, the sampling method used, the type of sample, and the total number of participants.
      • Where is the population from, and where was the study conducted? The limitations later mention that all participants are from one province, but no details on this are provided in the methods or text.
    • Sample Size Calculation: The tool used for calculating sample size is mentioned, but parentheses are left blank, and the reference is missing (line 80).
    • Exclusion Criteria: Some exclusion criteria are somewhat contradictory. The fourth criterion excludes participants who were not treated at home, while the fifth criterion excludes participants who were hospitalized for more than 7 days. Don’t these two criteria exclude each other? If we exclude participants who were not treated at home, do we also exclude those who were hospitalized for less than 7 days? This needs clarification.
    • Consent: It’s mentioned in the methods that surveys were conducted by phone and that participants consented before the study. However, at the end of the paper, it’s stated that all participants signed written informed consent. This also requires clarification. Did they consent by phone, or did they additionally sign informed consent?
    • Questionnaires: The methods section mentions two questionnaires were used. It’s not specified in which language they were administered—was it the original language or translated? Were the questionnaires validated for the population studied? If translated, what was the translation method?
    • DASS Questionnaire: Only the abbreviation “DASS” is provided without the full name. The full name of the questionnaire should be stated at least at its first mention.
    • Survey Responses: In line 150, it’s mentioned that participants “circled” their answers. It’s unclear how they circled answers if the survey was conducted by phone. A more suitable term would be “chose” an answer.
    • Metric Properties: In line 165, it states that the scale used has good “metric properties.” This is also unclear—exactly which scale is being referenced, as two questionnaires were used? For which population are these questionnaire properties applicable?
  2. Discussion:

    • The discussion feels somewhat fragmented as it consists of several small sections. It may be better to combine these into one section for coherence. Additionally, the discussion lacks conclusions and explanations. It primarily states the main results and compares them with other studies. Missing are potential explanations of the results and the significance of these findings, especially for the female population studied. What is the impact of this research, and what are the possible applications of its findings?
  3. Limitations:

    • Possible limitations of conducting a phone survey are not mentioned.
    • If the questionnaires were not validated for the studied population, this should also be mentioned among the limitations.
Comments on the Quality of English Language

The English could be improved to more clearly express the research.

Author Response

Dear Reviewer,

Thank you very much for contributing to the substantial improvement of the manuscript. 

Please find the detailed responses attached and the corresponding revisions/corrections highlighted in the re-submitted file.

Kind regards,

The authors

Reviewer 3 Report (New Reviewer)

Comments and Suggestions for Authors

The topic of the manuscript “The Impact of Physical Activity Level on Women's Mental Health After COVID-19 Infection” is interesting. However, the manuscript has several deficiencies that need to be corrected in order to be published in a scientific journal.

-Causal inferences should be avoided throughout the manuscript (including the title) (do not use the term "effects" or “impact”), as the study conducted does not allow for the establishment of cause-and-effect relationships.

-It is also suggested not to use the terms "pre-test" and "post-test" as they are confusing because both measurements were taken after COVID-19 infection and there was no intervention between the two measurements.

Abstract

-In the Abstract, lines 25 to 28, the following text appears “T-test was used to determine the differences in mean values between the initial and final measurements. To determine the impact of physical activity on mental health parameters, linear regression analysis was used. A p-value of <0.05 was set as the level of statistical significance for all statistical procedures”. It is suggested that this text be removed from the abstract as an abstract should be brief, clear and concise.

-On line 31 it says “higher MET values”. It should be specified what MET means since this is the first time it appears.

Introduction

-The introduction should be revised by clearly stating the specific problem under study. Since the study was conducted with women who have recovered from COVID-19, it should describe this situation, the problems it causes, and the possible mechanisms by which exercise is important.

-In the Introduction, lines 46 to 45, the following text appears “The World Health Organization (WHO) declared a global pandemic in March 2020, prompting widespread quarantine measures to limit the virus's spread [4]. These measures, while essential for controlling infections, led to significant mental health challenges, including increased anxiety, depression, and stress in both men and women”. The text "These measures, while essential for controlling infections, led to significant mental health challenges" should be changed, as it implies a cause-and-effect relationship between measures to limit the spread of the virus and mental health problems, but such mental health problems could also be due to other causes, such as, for example, fear of illness and fear of death for oneself or family members.

-On page 2, lines 70 and 71 it says “Therefore, the aim of this research is to iinvestigate the effects of physical activity on mental health parameters among women who have recovered from COVID-19”. The word “effects” should be replaced with “association” because, as mentioned above, the study presented does not allow causal assumptions to be made.

The hypothesis (page 2, lines 72 to 74) should be reformulated as it is very confusing.

 Material and Methods

-This section is very confusing. For example, on page 2, lines 78 to 70, it says “The population from which the subject sample was selected was defined as the population of women (n=190) aged between 20 and 60 (47.60 ± 11.1) who were infected with 79 COVID-19". It would be important to include information about what specific population is being referred to, including the specific dates and location where the study was conducted.

-The procedure for selecting participants, including the sampling method, should be clearly described.

-Inclusion and exclusion criteria are very confusing. Such criteria should be theoretically justified in the Introduction of the manuscript. Furthermore, on page 2, lines 67-68, it says “the impact of physical activity on the mental health of women after prolonged COVID-19 infection”; but in the inclusion criteria, line 92, it says "subjects hospitalized for up to seven days". ¿How long is considered to imply "prolonged COVID-19 infection"?

-The description of the International Physical Activity Questionnaire (IPAQ) should be revised as it is very confusing. For example, on page 3, lines 132 to 136, it says: "Using the IPAQ questionnaire, it is possible to assess the level of physical activity through various domains (physical activity at work; physical activity in transport; physical activity at home; physical activity in free time; physical activity and walking; moderate physical activity; intensive physical activity; overall physical activity), and intensities”.

-The description of the DASS questionnaire should also be revised and clarified. In addition, on line 149 it says "It contains 41 questions", but on lines 156-157 it says "Each of the three DASS scales contains 14 items". Therefore, the scale should contain 42 questions, not 41. In addition, information about the reliability of each scale in the sample of the present study should be included.

-The sociodemographic characteristics of the study sample should be presented.

Results

-Tables 1 and 2 should be revised to include some data on the effect size of the differences between the two assessments for each type of measure used. Also, in the last column of each table, the p-values ".000**" should be replaced with "<.001". And it is not necessary to include the text "Level of statistical significance * p< .05, level of statistical significance ** p< .01" in the legends of these tables.

-Lines 178-180 say "The paired samples t-test analysis revealed significant improvements in mental health parameters and physical activity levels between the initial and final measurements (Table 1). ". This text should be corrected because there are no data on physical activity in Table 1.

-In the Table 1 legend, lines 185-186 says "PA - physical activity; MET-metabolic equivalent ". This text should be deleted because there are no physical activity data in Table 1. In addition, the legend says "t-degrees of freedom”, but in this table the degrees of freedom do not appear

-In the legend of Table 2, lines 198-199, it says "The obtained values of mental health parameters are based on the DASS questionnaire”. This text should be deleted because there are no data on mental health in Table 2. In addition, the legend says “MET-metabolic equivalent; t-degrees of freedom”, but in this table there is no heading for "MET" and the degrees of freedom do not appear.

-In the first row of Table 3 (page 5) it says "Post-test", this text should be changed because the data presented in Table 3 are those of the initial measurement, not those of the post-test.

-On page 5, lines 217-218, the following text appears: "The predictor variables showing the most substantial impact on anxiety included overall physical activity (β = 19.39)". And on page 6, lines 219-220, it says: "For depression, the key predictors were overall physical activity (β = 15.70)”. This must be an error, since such results indicate greater depression and anxiety with greater physical activity. Perhaps the correct data are "-15.70" for depression and "-19.39" for anxiety. This should also be checked and corrected in the results in the last row of Table 4 "Overall PA" where the beta weights shown are "15.70" for depression and "19.39" for anxiety.

Discussion

The Discussion section should be revised by evaluating and interpreting the results obtained in a more integrated manner and in a way that is fully justified by the results obtained. An attempt should also be made to explain why exercise is not associated with depression or anxiety or stress in the first measure and why exercise is not associated with stress in the second measure.

Author Response

Dear Reviewer,

Thank you very much for contributing to the substantial improvement of the manuscript. 

Please find the detailed responses attached and the corresponding revisions/corrections highlighted in the re-submitted file.

Kind regards,

The authors

Round 2

Reviewer 3 Report (New Reviewer)

Comments and Suggestions for Authors

The revised manuscript, now entitled “Physical Activity Level and Women's Mental Health After COVID-19 Infection”, has been improved from the previous version. However, several minor errors have been identified which should be corrected and are listed below.

-Abstract: Line 36 says “COVID-19 subjects”. It should read “women” instead of “subjects”, as the study was conducted in women and it is not possible to know whether such results would also occur in men.

-In the introduction, page 2, line 50, it says “and family safety, and changes in daily life [treba da bude 10]”. What does “treba da bude” mean?

-In the introduction, on page 2, line 87, it says “Sample size was estimated using the pwr.t.test()”. What does “()” mean?

-On page 4, lines 190, 191 it says: “In terms of health status, most participants were non-smokers (71.1%), while 28.9% were 190 smokers.”. As health status is not the same as smoking behaviour, it is suggested that, for example, “smoking behaviour” should be used instead of “health status'” The same should be done in Table 1. Also, in Table 1 it says: “Martial status” it should say “marital” (not “martial”).

-On page 6, lines 223 and 224 it says: “low R² values 223 and high significance levels (all p > .05)”. It is suggested not to put “high significance levels” but to put, for example, “not statistically significant levels (all p > .05)”.

-As stated in the previous revision of the manuscript, if exact p values are given in the Tables, it is not necessary to put asterisks to indicate significance level, nor should the footnote text read “* p< .05, ** p< .01”. Table 2, Table 3, Table 4 and Table 5 should therefore be reviewed and corrected. Furthermore, as stated in the previous review, the p-values should not read ".000" but rather  "< .001" as it is not recommended to use any p-value smaller than .001.

Author Response

Dear Reviewer,

Thank you very much for your interest and contribution to the substantial improvement of the manuscript. 

Please find the detailed responses attached and the corresponding revisions/corrections highlighted in the re-submitted file.

Kind regards,

The authors

This manuscript is a resubmission of an earlier submission. The following is a list of the peer review reports and author responses from that submission.

Round 1

Reviewer 1 Report

Comments and Suggestions for Authors

In this manuscript, Stamenkovic et al., studied the relationship between the physical activity levels of women and their mental health after Covid-19 infection. They aim to determine whether the levels of physical activity were affected the mental health in the 14 weeks post-infection. They reported changes in physical activity after 14 weeks. The first measurement did not show any relationship between physical activity levels and mental health parameters. However after 14 weeks, the physical activity levels seemed to be associated with anxiety, depression but not stress.

The objectives described in the paper are consistent with studies showing that the global COVID-19 pandemic has impacted the mental health of individuals worldwide. However, despite a large number of participants recruited and a study that appears at first sight to be well conducted, there is a real weakness in the writing of the manuscript. Firstly, there is no part Discussion that is directly following the results in the same part. Writing must be improved. This paper seems to have a great potential with a substantial body of work and results. However it is not highlighted by the writing of this manuscript, which lacked clarity and conciseness. The material and methods part needs to be more precise as a lot of information are missing and the results part needs to be better organized, as we are lost in reading this part. You may need simplify the results part with only simple and efficient description. There is too many repetitions of several words, particularly the verb “to show”.

Abstract:

-        In the abstract, initial and final measurement are announced but we have no idea of what it is. It is after explained a little bit more in the M&M but clearly we need to understand 100% of the abstract

-        Also in the abstract, results are not well described: “T-test showed that there is a difference in the mean values between the initial and final measurements, ….”. It means unfortunately nothing, the changes are going in which way ? And after it seems that there is association between physical activity and mental health ? In which way ?

-        In the conclusion of the abstract, it is mentioned “prolonged infection of COVID-19”. Is it really or are you studying post-infection COVID-19 ?

Introduction:

-        You start by “the coronavirus”, here the real name of this virus needs to be written at least one and then it can be named COVID-19 for the manuscript

-        The introduction is not well written and the rationale is not clear, same for the objective. The objective is to determine whether there is an potential association or an effect of physical activity levels on mental health ? If it’s the second option, is a worst mental health could not lead to a decrease in physical activity or a increase ?

M&M:

-        Participants are the first describe, then the protocol design.

-        I did not understand the Physical Activity levels Questionnaire. And also, how to you calculate everything in your results after ?

-        Mental health questionnaire is more descriptive but what DASS means ? It is written “the scale has shown good metric properties”, this should be written somewhere maybe but not in the M&M

-        Is the normality of the data has been verified ? What are “basic statistical parameters “?

Results:

-        The writing of the results is really to long and does not describe well the results. This part must be only description. The writing must be improved.

-        The table 1 is concerning for several points. All the “p-value” that is written here “Sig” for no real reason (in other table too). SD are really important and there is still significant difference between initial and final measurement ? And all the p = 0.000 ? Mental Health and Physical Activity levels results must be separated in 2 tables. Finally,  how do you calculate PA at work, at home, in spare time, … ?

-        For the next part with the regression analysis, physical activity does not impact mental health or they do not seem related ? Beta and Sig., same as other table, usually they are not called by these names.

Comments on the Quality of English Language

However, despite a large number of participants recruited and a study that appears at first sight to be well conducted, there is a real weakness in the writing of the manuscript. Firstly, there is no part Discussion that is directly following the results in the same part. Writing must be improved. This paper seems to have a great potential with a substantial body of work and results. However it is not highlighted by the writing of this manuscript, which lacked clarity and conciseness. The material and methods part needs to be more precise as a lot of information are missing and the results part needs to be better organized, as we are lost in reading this part. You may need simplify the results part with only simple and efficient description. There is too many repetitions of several words, particularly the verb “to show”.

Author Response

Dear Reviewer,

Thank you very much for taking the time to review our manuscript, and for contributing to the substantial improvement of the manuscript.

Kind regards,

authors

Reviewer 2 Report

Comments and Suggestions for Authors

I appreciate your efforts to conduct research on a topic that has affected human health around the world in recent years. It is also noteworthy that you can even associate this situation with physical activity, which is becoming increasingly important. However, some improvements are needed in your research.

I wish you good work.

Abstract

Line 18-19: ‘’ The sample consisted of women (n=190) aged 20 to 60 (47.60±11.1) who were infected with COVID-19. ‘’

There are two notable situations here. First, how was this age group determined? Is there a special reason? Why is it not for people over the age of 18 or why wasn't a preference made based on menstrual cycle? The second issue is, are 190 people enough for such an ambitious research? Does it represent the universe you want to evaluate? It looks like your sample size is insufficient. A power analysis or alternative methods are needed to determine the correct number of samples. Please focus on this issue and provide necessary answers and edits.

Line 21-22: ‘’ T test was used to determine the differences in mean values between the initial and final measurements.’’

If this study is a pre-test post-test study type, let's write it with correct expressions. In addition, the research model that supports this should be stated in the method section.

Line 25: ‘’  initial and final measurements…….’’  Let's use correct expressions.

Introduction

Sufficient literature information is presented in the introduction section. However, based on this information, the original value of the research has not been presented well enough. The original value needs to be rewritten. In addition, every research starts with a hypothesis. However, this information is missing in your research. Please explain the main hypothesis of your research.

Materials and Methods

Please write and explain the research model and design.  Also offer calculation of sample size.

Has the Cronbach alpha value of the scale you use for mental health been calculated for your research? Prove that it is a reliable measure for your research results.

Data analysis

Was the normality of the distribution tested before performing statistical analyses? Based on what information were these analyzes chosen?

Please provide the necessary information.

Line 153: ‘’T-test was applied……’’ Which form was used?

Results

It will be healthier if the expression pre and post test is used in the tables.

F and R values ​​are given in the interpretation of Tables 2 and 3, but this information is not included in the tables. Please edit the tables.

Discussion

The discussion is presented as a continuation of the findings. Please separate it with an appropriate title.

Advantages and shortcomings of the study

Line 277-279: ‘’ The  advantage of this research is a large number of subjects are mostly higher educated people.’’

What advantage does this provide? It's not understandable. Please elaborate.

Author Response

(The authors gave the same response as above.)

Reviewer 3 Report

Comments and Suggestions for Authors

Dear Editor,

Thank you for the opportunity to review the paper entitled “The Influence of the Level of Physical Activity on Women's Mental Health After Covid-19 Infection”.  

In my opinion, I believe that the paper does not fill any gap in the literature because many researchers have examined the same factors after the covid-19 infection.

Thus, I believe that it has no particular value for publication and is unreliable due to its limitations.

In addition, the introduction section should incorporate a significant number of citations from the results.

Comments on the Quality of English Language

Minor editing

Author Response

(The authors gave the same response as above.)

Reviewer 4 Report

Comments and Suggestions for Authors

I hope this letter finds you well. I had the opportunity to review your article titled, “The influence of the level of physical activity on Women’s mental health after Covid-19 infection”, which was submitted Healthcare.

 1. Title

Prepositions are being used redundantly. I hope you reduce the use of Of.

 2. Abstract

Please remove Background, Methods, Results, and Conclusion. Please state the purpose of your research specifically. The research method is too specific. Please briefly describe the statistical techniques used in this study. The conclusion is so obvious. Based on the results found in this study, we hope to present conclusions that are different from those of previous studies.

 3. Introduction

This study is about the level of physical activity for women's mental health after COVID-19. However, this study only emphasizes the importance of explaining corona and physical activity. Why is there a perceived lack of need to study women's mental health? It is judged illogical to say that it is necessary simply because it has not been studied until now. In other words, it is judged that researchers are not clearly suggesting why women's mental health should be studied.

 4. Materials and Methods

It is necessary to explain why the ages of the research subjects were set to 20s and 60s. Please explain in detail the selection method for the 190 research subjects. It was impressive that the purpose and benefits of the study were explained to the subjects and their permission was obtained before the test began. The measurement tools, physical activity and mental health, are explained in detail.

 5. Results

It is judged that the results of the t test and regression analysis, which are statistical techniques established by the researcher to clarify the purpose of this study, were appropriately presented. However, statistical chances need to be corrected to italics. For example, p, t, sig, R, R2, and F. And I would like to change Overall PA (19.39) to PA(Beta=19.39). Please indicate the beta notation of the subsequent factors.

 Advantages and shortcomings of the study

I think the comparison between the research results and the prior research was very well done. However, it only lists previous studies similar to the results of this study, and does not present the researcher's argument. The researcher's argument regarding the results of this study would be explained by comparing it with previous studies, which would make the research better. The advantages and shortcomings of the study should be described after the conclusion.

 6. Discussion

There is no discussion in this study. I would like to discuss this in a new chapter. I would like to discuss this in a new chapter. Discussion is a very important part of research.

 7. Conclusion

I think the conclusion is the part that describes the purpose, method, and results of the study, as well as the uniqueness and academic value of this study. If this study were to describe the above content in more detail, it would be a better study. This conclusion is believed to be based on what has been revealed in previous studies.

It is necessary to revise it by referring to the academic society form. It was determined that the necessity, discussion, and conclusion of the study needed to be comprehensively revised.

Comments on the Quality of English Language

 Moderate editing of English language required.

Author Response

(The authors gave the same response as above.)
